# Biological Characteristics of a Rare and Vulnerable Species (*SCHIZOTHORAX ARGENTATUS* (Kessler, 1874)) of TOKYRAUYN RIVER and Approbation of Its Artificial Reproduction

Fariza Amirbekova [1,2], Kuanysh B. Isbekov [1], Saule Zh. Assylbekova [1], Olga A. Sharipova [1], Kamila Adyrbekova [1,3,*] and Nailya Bulavina [1]

[1] Fisheries Research and Production Center LLP, Almaty 100141, Kazakhstan; amirbekova@fishrpc.kz (F.A.); isbekov@fishrpc.kz (K.B.I.); assylbekova@fishrpc.kz (S.Z.A.); sharipova@fishrpc.kz (O.A.S.); bulavina@fishrpc.kz (N.B.)

[2] Department of Production Technology of Livestock Products, Faculty of Bioresources and Technology, Kazakh National Agrarian Research University, Almaty 050010, Kazakhstan

[3] Department of Molecular Biology and Genetics, Faculty of Biology and Biotechnology, Al-Farabi Kazakh National University, Almaty 050040, Kazakhstan

\* Correspondence: adyrbekova@fishrpc.kz

**Abstract:** As a result of the complex impact of anthropogenic factors on the ecosystem of Balkhash Lake, the main commercial species of fish—endemic—Balkhash marinka was on the verge of extinction. Artificial reproduction is becoming increasingly important every year in the complex of works to maintain the commercial stocks of valuable fish species as well as to preserve rare and endangered populations. Despite the fact that attempts at artificial reproduction have been repeatedly made, to date no scientific and methodological basis has been created to study and preserve rare and endangered fish species in the Republic of Kazakhstan and there are no adapted technologies for the formation of repair herds of fish of this category for the fish breeding enterprises of the reproductive complex of Kazakhstan. The presence of a replacement stock of the Balkhash marinka will further contribute to the development and improvement of biotechnical methods for their breeding and cultivation of viable stocking material. The stocking of natural water bodies with viable fish seeds and the maintenance of living collections of rare and endangered fish species will contribute to their conservation in natural conditions and, as a reserve, in the country's fish farms. The results of the development of effective technologies for artificial reproduction and the formation of a replacement stock of the Balkhash marinka in industrial conditions contribute to solving the fundamental problem associated with the preservation of rare and endemic fish species from complete extinction, genetic diversity and the rational use of the potential of natural populations. An analysis of genetic identification showed that the breeders of marinka, according to the results of sequencing the barcoding gene CO1 of mitochondrial DNA, belong to the species Balkhash marinka *S. argentatus* with an accuracy of 99–100%.

**Keywords:** reproduction; *Schizothorax argentatus*; reproductive products; larvae; hormonal stimulation; breeders; incubation; embryonic development

## 1. Introduction

There are three species of marinkas of the genus *Schizothorax* (Heckel, 1838) (*Teleostei: Cyprinidae*), in Kazakhstan: common type *S. intermedius* (Mc'Clelland) in the basin of Syr Darya River, Balkhash type *S. argentatus* (Kessler) in Ile-Balkhash basin and Ili type *S. argentatus pseudaksaiensis* (Herzenstein) in the River Ile [1,2].

The Balkhash marinka is an endemic species of ichthyofauna of the Ile-Balkhash basin. In the reservoirs of Central Asia, marinka do not have such a wide range of forms

characteristic of the Balkhash marinka. Before acclimatization work (1932–1952), the marinka in the catches of the eastern part of the reservoir occupied the second place [3]. In subsequent years, the regulation of Ile River flow on PRC territory, construction of the Kapchagai hydroelectric power station, change in the composition of the ichthyofauna in connection with the acclimatization of new species and pollution of the basin's water area with industrial and agricultural discharge waters became limiting factors for conservation and the sustainable existence of indigenous fish species [4].

Before the beginning of acclimatization work, the Balkhash marinka was a background species of the lake and rivers flowing into it. The range of the species was widespread in the waters of the Ile-Balkhash basin inhabiting all types of reservoirs—from mountain streams to large terminal reservoirs. Currently, a small population of the Balkhash marinka has been preserved only in the Tokyrauyn River; they are rarely met in other rivers flowing into the eastern part of Balkhash Lake. The Tokyrauyn is the only river of the northern Balkhash region in which a complex of indigenous species has been preserved.

The Tokyrauyn River originates from the southern slopes of the Berkara Mountains. The total length of the river from the source to Balkhash Lake is about 250 km. Tokyrauyn refers to the rivers of the so-called Kazakhstan type with a high short-term flood usually in the second half of April–early May, which accounts for up to 90–100% of the surface runoff, and a long period of low-water season [5,6].

In the context of increasing anthropogenic impact on water resources of the Ile-Balkhash basin, the artificial reproduction of fish is becoming increasingly important every year in maintaining the stocks of valuable species and preserving rare and endangered populations.

The technology of artificial reproduction of the Balkhash marinka is well developed, and specific measures to restore the number have been repeatedly introduced by specialists of the leading scientific institutions of Kazakhstan [7]. However, to date, no measures have been taken to restore natural populations. The Balkhash marinka is a species with a complex ecological structure of populations, so it performs a wide variety of functions in the ecosystems of water bodies. Consuming large quantities of submerged aquatic vegetation and organic residues (detritus), the Balkhash marinka will be indispensable in the conditions of increasing the eutrophication of the reservoirs of the Ili-Balkhash basin [8].

At the same time, it is important to note that artificial breeding measures should be accompanied by molecular genetic studies. The genetic structure of the species of Balkhash marinka *S. argentatus* is poorly understood, and there are no data at all in Kazakhstan. In global practice, genetic studies of various species of the genus *Schizothorax* using microsatellite markers and mitochondrial DNA are widely used by Chinese, Iranian and other scientists [9–11].

## 2. Materials and Methods

Biotechnical methods for creation of the Balkhash marinka replacement stock have been developed at the created fish breeding site of Balkhash Branch of FishRPC LLP. Department for growing juveniles was engaged from eggs' incubation using Weiss incubators to cultivation of juvenile marinka. In order to maintain Balkhash marinka breeders in prespawning period and conduct cultivation of juvenile and older age groups, a basin plot has been organized with a closed water supply system.

During expeditions in the period 2018–2020 on the Tokyrauyn River (Figure 1), the breeders were caught by fixed nets with cells of 20 to 45 mm in order to avoid injury, the nets were checked every 4 h and traps (creels) were used in shallow waters.

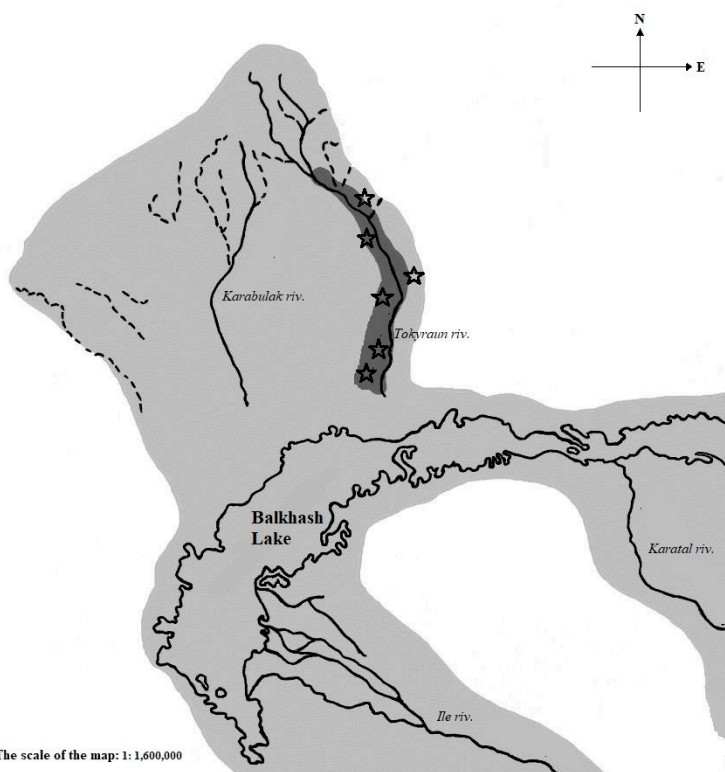

**Figure 1.** Sampling sites of Balkhash marinka in Tokyrauyn River (open pentagram represents the location of sampling sites).

The fish that were caught were transported by road in 120 L barrels with oxygen pumped into them for further placing in a recirculating aquaculture system. Delivery time from the natural habitat to the site did not exceed 2.0–2.5 h.

Analysis of the habitat by hydrochemical indicators of water was carried out in accordance with generally accepted methods and regulatory documents [12–17]. Compliance of the results of analyses with fishery standards was carried out according to the unified system for classification of water quality in water bodies [18].

Selection and processing of hydrobiological samples was carried out according to the generally accepted methodology [19]. Phytoplankton sampling was carried out with a single-liter bathometer by filtering 100 L of water and zooplankton sampling by Upstein's net. Phytoplankton samples were processed in Nageotte chamber and zooplankton samples in Bogorov chamber. Different groups of organisms were identified by corresponding determinants [20–25]. Macrozoobenthos was selected by Petersen's dredge. Organisms were determined by determinants [26–29]. The biomass of individual groups was determined by weighing on AP2140 electronic scale.

Sampling by analyzing the state of the Balkhash marinka was carried out from scientific catches. Processing of ichthyological material was carried out according to generally accepted methodology of Pravdin I.F., 1966 [30]. Age was determined by annual rings on scales with a microscopic method according to the generally accepted methodology [31,32].

Genetic material from the marinka of Tokyrauyn River which are protected as a species listed in the Red Book of the Republic of Kazakhstan was collected by the researchers from Balkhash Branch of Fisheries Research and Production Center from individuals caught as bycatch in the research fishing of Balkhash marinka.

Genomic DNA was extracted from pectoral fin tissues and then was fixed in 96% ethanol. A total of 28 specimens were examined from Tokyrauyn River.

DNA was isolated by column absorption (PALL) [33]. Analysis of mitochondrial DNA was carried out using the C1000/T100 (Bio-RAD, Hercules, CA, USA) Thermocycler.

Approximately 680 bp were amplified from the 5′ region of the COI gene using universal primers for CO1 [34]: FishF2_t1 TGT AAA ACG ACG GCC AGT CGA CTA ATC ATA AAG ATA TCG GCA C and FishR2_t1 CAG GAA ACA GCT ATG ACA CTT CAG GGT GAC CGA AGA ATC AGA A, respectively.

PCR reaction contained about 100 ng DNA and was run in a total volume of 15 μL (70 mM Tris–HCl (pH 8.3); 16.6 mM (NH4)2SO4, 2 or 3 mM MgCl2; 100 μL of each deoxyribonucleoside triphosphate; 1.5 μmol of each primer and 1 unit of Color Taq polymerase (Sileks)). The reaction conditions included initial DNA denaturation for 10 min at 95 °C, followed by 30 cycles of denaturation at 94 °C for 20 s; primer annealing at 52 °C for 40 s; primer extension at 72 °C for 60 s and final extension at 72 °C for 10 min. The results of amplification were tested by means of electrophoresis in 2% agarose gel with subsequent staining with ethidium bromide. Sequencing was carried out at both ends of the PCR of the product, i.e., two sequencing reactions were carried out to determine the nucleotide sequence. The resulting sequences (forward and inverse) were superimposed on each other to confirm the accuracy of reading the DNA of each sample. The sequencing reaction was carried out in amplifier C1000/T100 (Bio-RAD) or similar in accordance with the instructions for the device.

Sequencing was performed with the same primers on a Nanophore 05 [35] sequencer with a BigDye™ TerminatorKit v.1.1/3.1 (Applied Biosystems, Waltham, MA, USA). Further analysis and alignment of the obtained sequences was carried out with the Geneious bioinformatics software package. A haplotype network was built with software program [36].

Determination of species affiliation of the tested sample was conducted by comparing the obtained sequence with reference sequences of various fish species deposited in the international database of sequences of mitochondrial DNA control region by the method of searching for paired matches (BLAST). For this purpose, we used open database NCBI—National Center for Biotechnology Information (http://www.ncbi.nlm.nih.gov, accessed on 26 July 2022).

Hormonal stimulation of spawning of breeders in artificial conditions and natural environment was carried out under recommendations [37–39]. Hormonal stimulation to producers of the Balkhash marinka was carried out with acetonated pituitary glands of cyprinids collected during the period of prespawning migration of fish previously treated with acetone and dried. This method is considered classical and is used in the reproduction of carp, grass carp and silver carp in the conditions of fish farms.

Females of the Balkhash marinka were applied a threefold scheme of injection by carp pituitary. The first (preliminary) dose of the drug to females was 0.3 mg. Choice of such a dose ensures progressive maturation of eggs with current hormonal background. The second dose in the amount of 0.3 mg/kg was administered after 12 h. Control over injected females showed that administered dose of the permissive injection was not sufficient to obtain mature reproductive products, and after 36 h it was additionally injected (pituitary solution) in an amount of 2.0 mg. At the same time, hormone therapy was carried out to male breeders with 2.0 mg dose.

In accordance with the readiness of breeders to spawn, lifetime sampling was carried out by manual decanting. For degumming of eggs, low-fat milk was used in a concentration of 1:10 (1 L of cow's milk per 10 L of water). Incubation of fertilized eggs was carried out at fish breeding site in the Weiss incubators and in the natural environment—in Ses-Green apparatuses [40].

Embryonic development was examined and controlled using a stereoscopic microscope MSP-1 and a digital USB microscope [41].

Behavior and condition of the fish were controlled and examined every day in order to identify diseases. When feeding the larvae and fry of the Balkhash marinka obtained in industrial conditions, we used live food (decapsulated eggs of the Artemia crustacean, bloodworm, cyclops and daphnia) and starter feeds of Aller Aqua Danish Company produced in Poland. When growing fry, feeding was carried out around the clock every two to three hours, and the amount of feed and frequency of feeding were calculated taking into account

physiological state of the fish, changes in temperature and oxygen regimes [42]. The daily rate of feeding fingerlings was 4–6% of body weight, the multiplicity was 5 times, annuals were fed 4 times a day, and the amount of feed was 2–3% of the ichthyomass. In the feeding of yearlings, SUPREME production feeds of Alltech Coppens (Germany) were used.

Control weighing of marinka fry was carried out every 10 days on electronic scales with an accuracy of 0.01 g and of fingerlings and yearlings once a month.

## 3. Results and Discussion

The Tokyrauyn River is located in the northern part of the Balkhash Lake and is the largest watercourse here. The total area of the river basin is: below the mouth of Zhinishke River—6420 km$^2$; near the village of Narmanbet—7800 km$^2$; in the area of the mouth (together with Kussak River basin)—more than 23,000 km$^2$. However, most of the basin area (about 17,000 km$^2$ or 74%) is not involved in the formation of the flow of the main river (Tokyrauyn) and the actual value of the catchment is not more than 6000 km$^2$. The largest feeders are Rivers Zhalanash, Karamendy, Segizbay (Kassabay), Karatal and Zhinishke. Although the Kussak River has a basin of more than 10,000 km$^2$ (more than the main river), it is lost and does not reach the Tokyrauyn River. The average annual runoff of the Tokyrauyn River near Aktogay village is 78.0 million m$^3$ and below the mouth of the Zhinishke River is about 62.0 million m$^3$. At the mouth of the river (Baktai tract), the flow occurs only in high-water years during the flood period. In such years, the water sometimes reaches the lake. The size of the river flow is characterized by high variability in the interannual aspect. The largest discharge of water over the past 40 years has been more than 500 m$^3$. Like all rivers in Kazakhstan, more than 90% of the annual runoff of the river near Akotogay village occurs in spring during the flood. While moving down, the share of spring runoff increases and reaches 100% in the lower reaches. Moreover, in the area of the Baktai tract, the runoff occurs once every 10–15 years.

In 2016–2018, the flood at the mouth of the Tokyrauyn River was very high: water spills reached 100–200 m and in places it was wider at a depth up to 1.0–1.5 m.

To date, the Tokyrauyn River has a general trend: it has a low-water period which leads to a decrease in water.

In the spring-summer of 2021, the depths at sampling points varied between 0.5 and 2.5 m. The water transparency was determined almost to the bottom. In early May, the temperature of water masses reached 9.4–9.7 °C, and in the third decade of the month the water warmed up slightly to the temperature values of 10.6–12.2 °C. In August, the water temperature corresponded to 18.0 °C.

The hydrochemical characteristics of the Tokyrauyn River are based on the studies conducted in 2018–2021 (Table 1).

According to the sum of dissolved salts, the water was fresh; the mineralization this year was 762–774 mg/dm$^3$. According to dominant ions, the water belonged to bicarbonate class, calcium group, type II [43]. In the interannual dynamics, there was an increase in salt-forming ions and the mineralization of water.

The aquatic environment of the Tokyrauyn River is favorable for the vital activity of hydrobionts and, in terms of its hydrochemical characteristics, meets the requirements for water bodies of fishery importance.

In 2021, only six species of representatives from diatoms were identified in the taxonomic composition of phytoplankton of the Tokyrauyn River. The number constituted 120 mln.cl/m$^3$ with 0.643 g/m$^3$ biomass. *Amphora ovalis* Kützing (0.227 g/m$^3$) and *Epithemia turgida* (Ehrenberg) Kützing (0.329 g/m$^3$) dominated in the biomass. Based on the biomass size of phytoplankton under the trophicity scale, the River Tokyrauyn can be attributed to a low-class oligotrophic type [44]. In comparison with the studies of 2018, productivity decreased from moderate (1.063 g/m$^3$) to low (0.643 g/m$^3$), which is possibly associated with sampling on river flow, as a result of which the algae were removed, and it affected the floral density and productivity of the phytoplankton.

**Table 1.** Hydrochemical indicators of water of the Tokyrauyn River (average values).

| Indicators | Unit | 2018–2020 | 2021 | Quality Standard of Water Classes 2 and 3 |
|---|---|---|---|---|
| pH | - | 7.83 | 7.91 | 6.5–8.5 |
| Dissolved oxygen | $(mg/dm^3)$ | 7.3–7.9 | 8.7 | not less than 6.0 |
| Carbon dioxide | $(mg/dm^3)$ | 7.9–13.2 | 24.2 | 44.0 |
| Permanganate index | $(mgO/dm^3)$ | 2.6–3.2 | 4.0 | 10.0 |
| Ammonium nitrogen | $(mg/dm^3)$ | 0.06–0.11 | 0.10 | 0.5 |
| Nitrites | $(mg/dm^3)$ | 0.002–0.003 | 0.001 | 3.3 |
| Nitrates | $(mg/dm^3)$ | 1.25–1.58 | 0.60 | 45.0 |
| Phosphates | $(mg/dm^3)$ | 0.006–0.009 | 0.004 | 0.2 |
| Total iron | $(mg/dm^3)$ | 0.01–0.02 | 0.01 | 0.3 |
| Overall stiffness | $(mg\text{-}eq/dm^3)$ | 5.40–5.98 | 6.82 | – |
| Calcium | $(mg/dm^3)$ | 56.9–81.8 | 90.2 | not rated |
| Magnesium | $(mg/dm^3)$ | 22.8–34.0 | 28.3 | 30 |
| Sodium + potassium | $(mg/dm^3)$ | 78.0–95.0 | 95.8 | – |
| Bicarbonates | $(mg/dm^3)$ | 271–305 | 310 | not rated |
| Chlorides | $(mg/dm^3)$ | 61.7–64.6 | 76.6 | 350 |
| Sulphates | $(mg/dm^3)$ | 110–149 | 156 | 250 |
| Mineralization | $(mg/dm^3)$ | 619–711 | 768 | 1000 |

The zooplankton of the Tokyrauyn River in 2021 were represented by 13 species from three main groups—10 rotifers (*Notommata pseudocerberus* (Beauchamp), *Epiphanes macroura* (Barrois & Daday), *Trichotria truncata truncata* (Whitelegge), *T. pocillumpocillum* (Muller), *Mytilinaventra lisventralis* (Ehrenberg), *Euchlanis incise* (Carlin), *E. dilatata dilatata* (Leydig), *E. lyralyra* (Hudson), *Keratella quadrata quadrata* (Muller) and *Notholca acuminata acuminata* (Ehrenberg)), and 3 Cladocera (*Acroperu sharpae* (Baird), *Chydorus sphaericus* (O.F.Muller) and *Alonella nana* (Baird)); of the copepods, only the younger age stages were present—nauplius and copepodites.

In the biomass, copepods—46%—dominated. Rotifers and cladocera participated in equal shares—27% each in the formation of the river biomass. In addition to true zooplankters, ostracoda and insect larvae were detected in the sample. The numerical dominants of the river were rotifers with 60%, subdominated by copepods with 30% (Table 2).

**Table 2.** Numerical development of zooplankton in the Tokyrauyn River (average values).

| Major Groups | Number, Thousand Copies/$m^3$ | | Biomass, g/$m^3$ | |
|---|---|---|---|---|
| | 2018 | 2021 | 2018 | 2021 |
| Rotifers | 0.350 | 2.20 | 0.001 | 0.003 |
| Cladocera | 0.020 | 0.40 | <0.001 | 0.003 |
| Copepods | - | 1.10 | - | 0.005 |
| Total | 0.370 | 3.70 | 0.001 | 0.011 |

The species composition of zooplankton in 2018 was also composed of 13 taxa where rotifers predominated—85% of which formed the basis of abundance and biomass. Due to its peculiarity, the zooplankton of rivers is always poor and enriched by the removal of organisms from dead stream branches and backwaters. As the least agile swimmers, rotifers are primarily carried out into the river occupying a prevailing position not only in terms of species but also frequently in quantitative indicators. Moreover, although the production indicators for 2021 were rather high comparing with 2018, the bioproductivity

of the Tokyrauyn River had not changed and corresponded to the α-oligotrophic type with a very low feed class [44].

The bentofauna of the River Tokyrauyn in 2021 were represented by six taxa: gastropods, *Valvata planorbulina* (Paladilhe, 1867); ephemeroptera, *Caenis macrura* (Stephens, 1835); chironomid larvae, *Cryptochironomus gp.viridulus* (Fabricius, 1805), *Paratendipes gp.albimanus* (Meigen, 1818) and *Cladotanytarsus gp.mancus* (Walker, 1856), and woodlice, *Serromyia* (Meigen, 1818). The main abundance and biomass were chironomid larvae at 82.4% and 93.6% (Table 3).

**Table 3.** Quantitative development of zoobenthos in the Tokyrauyn River (average values).

| Group of Organisms | Population pcs/m$^2$ | | Biomass, g/m$^2$ | |
|---|---|---|---|---|
| | 2018 | 2021 | 2018 | 2021 |
| Oligochaetes | 30 | - | 0.01 | - |
| Nematodes | 60 | - | 0.02 | - |
| Pincers | 100 | | 0.03 | - |
| Bugs | 70 | - | 0.04 | - |
| Clams | - | 40 | - | 0.5 |
| Ephemoptera | - | 320 | - | 0.2 |
| Chironomid larvae | 120 | 2240 | 0.06 | 11.7 |
| Caddisfly larvae | 70 | - | 0.05 | - |
| Beetle larvae | 10 | - | 0.01 | - |
| Heleid larvae | 120 | - | 0.06 | - |
| Biting midge | - | 120 | - | 0.1 |
| Total | 510 | 2720 | 0.25 | 12.5 |

It should be noted that, in 2018, macrozoobenthos was represented by worms—oligochaetes and nematodes—and aquatic insects—ticks, bugs and insect larvae. Chironomid larvae also dominated in quantitative indicators (24%). Freshwater ticks (according to the number 19.6%) and caddis larvae (20% in biomass) were subdominated in quantitative development. In general, along the river, the quantitative development of macrozoobenthos is 510 pcs/m$^2$ and 0.25 g/m$^2$. Based on the biomass of zoobenthos, the Tokyrauyn River belongs to the ultraoligotrophic type of reservoir with the lowest class of feeding.

Currently, there is an increase in the quantitative indicators of zoobenthos: the number is 2750 pcs/m$^2$ and the biomass is 12.5 g/m$^2$. According to the generally accepted classification [44], the trophicity level of the River Tokyrauyn corresponds to an elevated class of the α-eutrophic type.

Balkhash marinka is characterized by an unpretentiousness to food objects; its ration includes both animal and plant organisms. This fact should be considered while developing feeding technology in order to domesticate the marinka in industrial conditions.

Originally, the Balkhash marinka was widespread in the Ile-Balkhash basin inhabiting all types of reservoirs—from mountain streams to large terminal reservoirs. The studies conducted in 2018–2021 showed that a fairly large reserve of Balkhash marinka was preserved only in the upper and middle parts of the Tokyrauyn River in the Northern Balkhash region.

Ready alignments were imported to Geneious Software [36], where the first neighbor-joining trees were prepared to check the datasets for potential sample mix-ups. In Geneious Software, the numbers of variable and parsimony-informative positions were obtained and the genetic differences between the lineages were counted. To allow easy comparison and comparability with other studies, we used the proportional genetic difference converted into percentages. The phylogenetic analyses of the COI gene sequences (28 totals) of the Balkhash marinka (A) show a divergence into Ili marinka clades (B). The phylogenetic analyses were performed using the Bayesian inference (BI) and maximum likelihood (ML).

An mtDNA CO1 gene fragment of 680 bp in size was obtained in 28 Balkhash marinka specimens. A total of two mitochondrial haplotypes typical of Balkhash marinka and Ile were identified. One of these haplotypes was completely consistent with the only previously published sequence for this species (GenBank ID 20356470). As can be seen from the figure, the Balkhash-Ili population was characterized by the well-separable haplogroups (Figure 2).

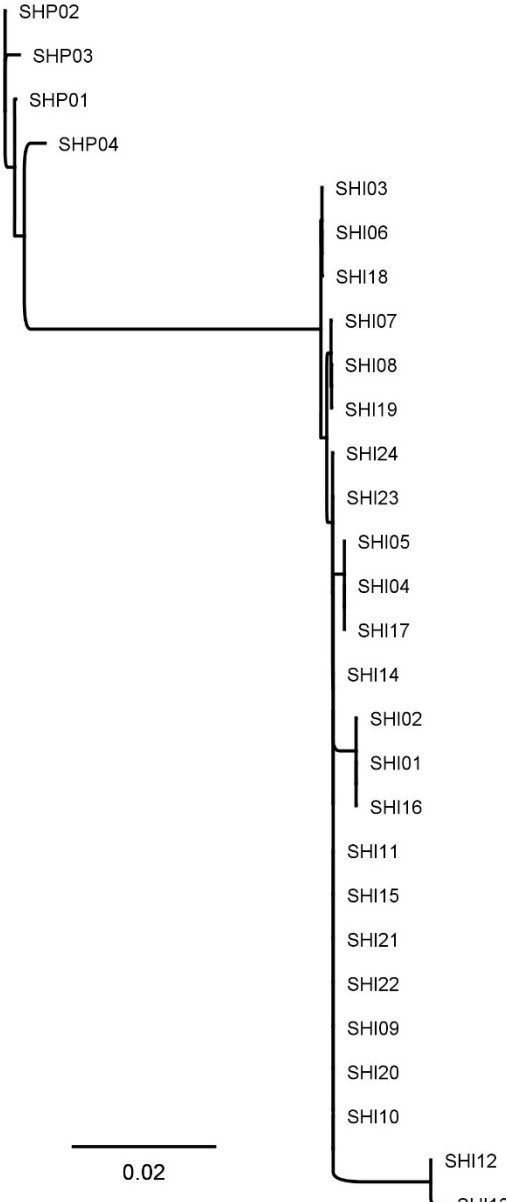

**Figure 2.** Neighbor-joining tree of marinka from DNA barcode sequences.

The results of the barcoding of the CO1 gene of the mitochondrial DNA showed that specimens caught in the Tokyrauyn River belong to the species Balkhash Marinka *S. argentatus* (SHI01-24) For comparison, the Ili Marinka—*S. pseudoaksaiens* (SHP01-04)—is shown as an outgroup.

The age variability of morphological features is expressed in the fact that the body height of immature specimens is greater, the head is bigger and the fins are more developed with a short body and long tail [45]. The biological characteristics and age composition of the Balkhash marinka from the Tokyrauyn River are provided in Table 4.

**Table 4.** Biological characteristics and age composition of the Balkhash marinka from Tokyrauyn River.

| Age, Year | Length (cm) | | Weight (g) | | Fulton's Condition Factor (Average%) |
|---|---|---|---|---|---|
| | Min–Max | Average | Min–Max | Average | |
| 2+ | 14 | 14.0 | 40–42 | 41 | 1.49 |
| 4+ | 21–23 | 22.2 | 128–188 | 155 | 1.41 |
| 5+ | 24–26 | 25.3 | 186–252 | 224 | 1.38 |
| 6+ | 26–28 | 27.1 | 252–362 | 292 | 1.49 |
| 7+ | 29–32 | 30.2 | 350–554 | 418 | 1.52 |
| 8+ | 32–36 | 33.8 | 458–690 | 565 | 1.48 |
| 9+ | 36–37 | 36.7 | 728–861 | 757 | 1.44 |
| 10+ | 38–41 | 39.8 | 862–1105 | 957 | 1.50 |
| 11+ | 42–47 | 44.0 | 1164–1256 | 1210 | 1.51 |

The maximum weight of the marinka was 3.5 kg in the 50–60 s of the last century, and the greatest length constituted 59.5 cm. The maximum observed age of males was 20+ and females 22+. To date, in scientific catches, the largest was an 11-year-old specimen (sexually mature female) with 47.0 cm length weighing 1.256 g. The rate of the linear growth and the condition of the Balkhash marinka were average. The presence of fingerlings-yearlings and second summer specimens with 5–11 cm body length and 5–20 g weight indicates the fact that the Tokyrauyn River has a self-reproduced stock of the Balkhash marinka. It has a sexual dimorphism: females are larger than males. Visually, they can be distinguished by a longer fin that goes beyond the middle of the tail stem and reaches the base of the tail fin. The catches conducted in 2021 were dominated by females; the sex ratio of males/females in spring was 2/3 and in summer was 1/3. In the Figure 3. Balkhash Marinka *S. argentatus*: the head from the side and full length of the body.

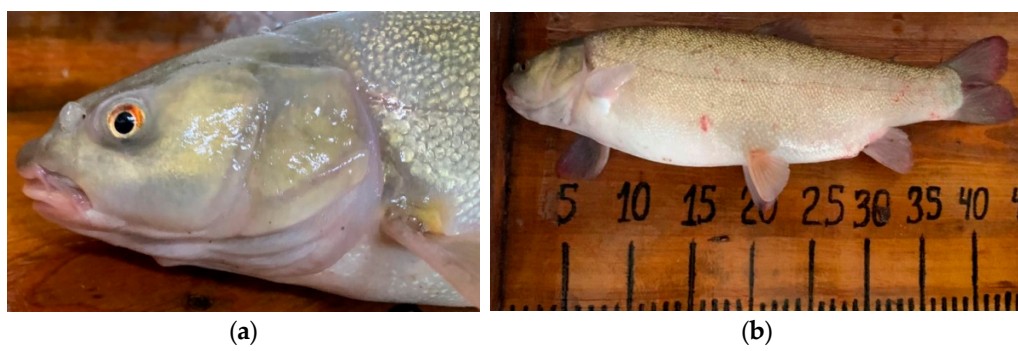

(**a**)　　　　　　　　　　　　　　　　　　　　　(**b**)

**Figure 3.** Balkhash Marinka *S. argentatus*: (**a**) the head from the side; (**b**) full length of the body.

The Balkhash marinka belongs to the group of fish with a single-batch spawning [45]. The first sexually mature males are found at three years of age and the first sexually mature females in a four-year-old. The main part of males matures at four and five years and females at five and six years. According to our research, spawning in the Tokyrauyn River takes place from the middle of April to the second decade of May at a temperature from 9 to 11 °C and above. In early May 2021, in catches, spawning females accounted for 70% of sexually mature specimens and 30% were females with reproductive products at maturity stage IV-V. According to bioanalysis results, sexually mature males were characterized by fluidity and the VI-II stage of maturity of sexual gonads.

The fertility of the marinka depends on its age, length and weight. Absolute fecundity ranges from 16.1 to 67.9 thousand eggs. Marinka lay eggs on river shoals and stony ponds at a depth of no more than 1–2 m. The body color of the Balkhash marinka is silvery-white

for immature specimens; the color of the large specimens varies greatly depending on the habitat: from dark silver to dark olive, reaching black.

Incubation was carried out both at hatcheries and in the natural habitat of the Balkhash marinka—in the Tokyrauyn River. In order to obtain reproductive products, sexually mature specimens were selected (Table 5).

**Table 5.** Fish and biological indicators of breeders of Balkhash marinka in natural habitat.

| Parameters of Breeders | Female Marinka | Male Marinka |
| --- | --- | --- |
| Number of fish (pcs) | 3 | 3 |
| Age of fish (year) | 8–10 | 7–9 |
| Weight (g) | 780–1050 | 330–440 |
| Fatness condition coefficient (%) | 1.4–1.5 | |
| Maturity coefficient (%) | 9.6–11.1 | 7.5–10.2 |
| Working fertility (thousand pieces) | 10.650–41.340 | |
| Relative fertility (thousand pieces/kg) | 13.420–46.535 | |
| Diameter of mature eggs (mm) | 1.8–2.4 | |
| Fertilization (%) | 90 | |

The scheme of hormone therapy depends on the stage of maturity of the gonads of individuals as well as temperature regime. In the range of spawning temperatures, to females with a high degree of readiness for spawning (IV maturity stage close to V), the drug was administered twice at intervals of 12 h. Using the biopsy method, it can be determined that the nuclei of most of the oocytes of the older generation are located at the shell. The amount of the drug for preliminary injection is usually 10% of the total dose. With an increase in temperature, the dose of gonadotropic material should be lowered.

The stage of egg maturation and readiness for spawning was stimulated by pituitary injections. Given the initial state of the females and the low temperature regime in the pools of the fish breeding area (16.6–17.8 °C), a three-fold injection scheme with an acetonated carp pituitary gland was applied to them. The first (preliminary) dose of the drug to females was 0.3 mg per kg of body weight. The choice of such a small dose ensures the progressive nature of the process of the maturation of the eggs with the existing hormonal background. Permissive injection was performed after 12 h. An observation of the state of the injected females showed that the injected dose of the permissive injection was not sufficient to obtain mature reproductive products, and after 36 h the drug was additionally administered in an amount of 2.0 mg/kg (Table 6).

**Table 6.** Timing of hormonal stimulation and recommended doses of carp pituitary solution used to stimulate marinka producers.

| Venue | Time between Injections (hours) | | | Time of Ovulation of Mature Eggs after the Last Injection (hours) |
| --- | --- | --- | --- | --- |
| | Preliminary (mg/kg) | Permissive (mg/kg) | Second Permissive (mg/kg) | |
| Tokyrauyn River | 12 (0.3) | 12 (3.0) | 12 (2.0) | 11–12 |
| Incubation manufactory of recycling aquaculture system | 12 (0.3) | 12 (3.0) | 12 (2.0) | 28–32 |

In obtaining artificial eggs, it is important to know the period of eggs' maturation; otherwise, females spawn on their own. Every 2–3 h, females were examined for the maturation of reproductive products. An enlarged and soft abdomen is considered a sign of ovulation. Eggs can be felt with careful palpation while the genitourinary tubercle protrudes by 1–2 cm. The ovulation of eggs in the females of the Balkhash marinka after

the last injection occurred on the Tokyrauyn River after 11–12 h and at the hatchery after 28–32 h.

Considering the readiness of breeders to spawn, lifetime sampling was conducted by manual decanting. The eggs were carefully squeezed out of the lower abdomen so as not to damage the internal organs of the fish. The time for decanting eggs and sperm before mixing them did not exceed 5–10 min.

The duration of insemination for the Balkhash marinka was 3 min. Sperm were added to the eggs and mixed gently with feathers for 2 min to increase sperm activity, and water was added twice at intervals of 20 s. The eggs of the marinka were large, light yellow in color, with high stickiness. A degumming procedure was carried out for 40 min with constant stirring in a solution.

The duration of incubation depends on the temperature of water and amount of oxygen. During the incubation period in the Weiss incubators, when the water temperature was 18.2–18.4 °C, the dissolved oxygen content was optimal—8.8–8.9 mg/dm$^3$ (95–96% saturation) (Table 7).

**Table 7.** Abiotic parameters during incubation of Balkhash marinka eggs in industrial conditions.

| Arrangement | Water Temperature, °C | pH | Oxygen | |
| --- | --- | --- | --- | --- |
| | | | mg/dm$^3$ | % Saturation |
| Fertilization and initiation of incubation | 18.2–18.4 | 8.21 | 8.8–8.9 | 95–96 |
| Pecking of larvae | 18.7 | 8.21 | 8.3–8.7 | 97–100 |

Duration of egg incubation was 5 days which, at an average temperature of 18.3 °C, corresponds to 91 degree-days.

It is also necessary to monitor the water flow. During the first 10 h, the eggs moved slowly, and the water flow was 0.6–0.8 L/min. Then, eggs were moved frequently, and the water flow was at the level of 1.0–1.2 L/min. The water supply should be adjusted so that the eggs are in a calm state but do not settle. Up to 35 thousand eggs of the Balkhash marinka can be loaded into one Weiss incubator with a volume of 8 L. The regular inspection of egg development included checking and adjusting the flow of water, removing infertile or dead eggs and applying the fungicide in a timely manner in case of fungal diseases.

By the embryonic period, we mean a period of time characterized by endogenous nutrition and covering development from the moment of fertilization to the beginning of the assimilation of food captured from the outside (until transitioned to mixed nutrition). The duration of the embryonic period is influenced by abiotic (temperature and gas regime) and biotic (yolk sac reserves, food security during the transition to mixed nutrition and biological characteristics of breeders) factors.

Microscopic observations of embryonic development should be carried out constantly during the entire incubation period, which will allow for assessing the critical stages of embryonic development, adjusting the conduct of certain technological operations and controlling the percentage of eggs with impaired embryonic development. The determination of the stages of early ontogenesis was carried out according to the guidelines [46].

After entering the water, the eggs begin to swell, which is associated with the formation of the perivitelline space (stage I). The cytoplasm is drawn to the animal pole and forms a tubercle—a blastodisc (stage II). At a water temperature of 18.2 °C, the duration of the stage was about 2 h. Then, there are stages of cleavage and blastulation, the duration of which at a temperature of 18.3 °C was 26–28 h.

At stages III, IV and V of the embryonic—gastrulation and organogenesis—fouling of the yolk sac, the formation of eye bladders and rudiment of auditory placodes were observed. The segmentation of the trunk began at the age of 3 days 2–5 h at a water temperature of 18.3 °C.

The next stages VI and VII include the separation of the tail section from the yolk sac (embryo at the age of 3 days 10–12 h). At this stage, the tail bud lengthens and forms the embryo tail section. It differentiates the chord, spinal cord and somites. The essential point

of this stage is the formation of the heart. It develops from mesodermal cells that separate in front of the right and left lateral plates, at the junction.

At stage VIII, the gill–jaw apparatus (development outside the shell) is developed and larvae hatch from the eggs, which is facilitated by the active movements of the embryos and a decrease in the strength of the eggs' shells. During this stage, there is a rapid development of the jaw and gill apparatus as well as other organ systems that prepare the body for an active lifestyle. In our case, the pecking of embryos from the shell (free embryos) began on the fifth day. The prelarvae were born with a large yolk sac, and the oral apparatus was not developed. The pecked embryos reached a length of 6.7–6.9 mm. There was no pigment on the body; the larvae were transparent.

The individual stages of the embryonic period of the life cycle of the Balkhash marinka are reflected in Figure 4.

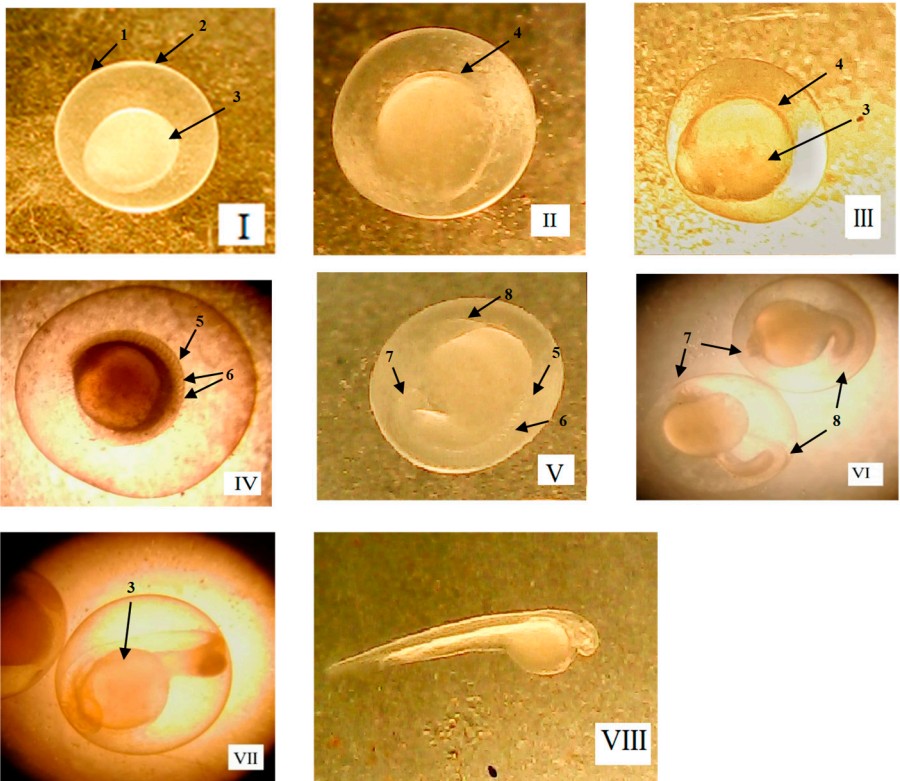

**Figure 4.** Stages of the embryonic period of the life cycle of the Balkhash marinka; stages (**I–III**)—activation of the egg and formation of the blastodisc; stage (**IV**)—gastrulation; stage (**V**)—organogenesis; stages (**VI,VII**)—separation of the tail section from the yolk sac; stage (**VIII**)—postembryo of the Balkhash marinka and development of the gill–jaw apparatus. Arrows: 1—caviar shell; 2—perivitelline space; 3—yolk sac; 4—blastodisc; 5—neural tube; 6—somites; 7—the head section of the embryo; 8—tail section of the embryo.

The temperature regime for prelarvae preservation corresponded to 18.7–19.0 °C, and the content of dissolved oxygen was high—8.3–8.7 mg/dm$^3$ (97–100% saturation). On the fifth day, the larvae developed swim bladders at a temperature of 19.0 °C. On the eighth day, the larvae switched to a mixed diet, and the embryonic period of development of the Balkhash marinka ended.

No special care is required during prelarvae preservation, and it is necessary to remove empty shells from eggs and dead larvae, monitor the hydrochemical regime and carefully monitor sanitary conditions in the tanks.

An analysis of embryonic development showed that fertilization on a natural reservoir was 90% and in artificial conditions was 80%. The obtained high result indicates properly selected biotechnical methods of the induced ovulation and fertilization of eggs.

## 4. Conclusions

Increased anthropogenic load on water bodies in the second half of the 20th century led to a sharp reduction in stocks and a decrease in the natural reproduction of fish of the Ile-Balkhash basin. A serious restructuring in ecosystems occurred as a result of large-scale acclimatization activities in 50–60 years.

Before acclimatization work (1932–1952), the marinka occupied one of the leading places in the catches of the eastern part of the reservoir. In subsequent years, the regulation of Ile River flow on PRC territory, construction of the Kapchagai hydroelectric power station, change in composition of ichthyofauna due to the acclimatization of new species and pollution of the basin's water area with industrial and agricultural discharge waters became limiting factors for the preservation and sustainable existence of indigenous fish species, in particular the Balkhash marinka.

An analysis of genetic identification showed that the individuals from the Tokyrauyn River according to the results of the sequencing of the barcoding gene CO1 of mitochondrial DNA belong to the species of Balkhash marinka *S. argentatus* with an accuracy of 99–100% and are clearly separated from the Ili marinkas.

Currently, artificial reproduction is recognized as one of the main ways to preserve rare and vulnerable species of fish. The presence of the self-reproduced stock of this endemic species in the river served as an impetus for fish farming work on domestication and the artificial breeding of the Balkhash marinka in industrial conditions.

In Kazakhstan, fish breeding organizations and hatcheries do not engage in the reproduction of rare and vulnerable species of fish. To date, technologies for obtaining and growing rare species of fish have not been developed. In this regard, the preservation of the Balkhash marinka can be solved only by performing the following measures: the formation of a broodstock; the development of the artificial production of the reproductive products and incubation of eggs and the development of biotechnology for cultivating the fish seeds in order to stock the water bodies.

As a result of this work, the data on habitat and spawning conditions of the marinka have been analyzed, similar conditions have been created in a recirculating aquaculture system and the marinka breeders were domesticated and adapted to artificial conditions. The technological methods of the reproduction of marinka in artificial conditions, hormonal stimulation and incubation measures have also been developed, the positive results of which have shown the fundamental possibility of reproducing the Balkhash marinka in industrial conditions. In addition, we have analyzed the data of the embryological development of the marinka in early stages and practically confirmed the high level of ovulated marinka eggs in artificial conditions. For the first time, a scheme for the hormonal stimulation of marinka was developed in order to obtain reproductive products, reproductive products were obtained, the fertilization and incubation of fertilized marinka eggs were carried out in a recirculating aquaculture system and standards for the reproduction of the marinka in recirculating aquaculture systems were developed. We continue to work in this direction and cultivate our own stocking material in a recirculating aquaculture system in order to form own broodstock by the method "from eggs" and methods to cultivate breeders in the natural reservoir. The ultimate goal is to obtain viable juvenile Balkhash marinka by the artificial insemination and incubation of eggs for the further cultivation and formation of the broodstock in industrial installations as well as the stocking of native reservoirs.

**Author Contributions:** Data Curation, F.A.; Supervision, K.B.I.; Project Administration, S.Z.A.; Resources, O.A.S.; Writing—Original Draft Preparation, K.A.; Methodology, N.B. All authors have read and agreed to the published version of the manuscript.

**Funding:** The conducted research of the genetic identification of the declared species of the marinka showed that all individuals cultivated on the Tokyrauyn River, in order to obtain the broodstock and further reproduction, belong to the species Balkhash marinka *S. argentatus* according to the results of sequencing the barcoding gene CO1 of mitochondrial DNA. The results obtained could facilitate

the development of standards for obtaining the marinka broodstock and artificial reproduction at the reproduction complexes of the Republic of Kazakhstan and further on carry out the annual reproduction and stocking of the Balkhash marinka in natural reservoirs to preserve a rare and vulnerable species of fish. The research is funded by the Ministry of Ecology, Geology and Natural Resources of the Republic of Kazakhstan (Grant No. BR10264236).

**Institutional Review Board Statement:** Not applicable.

**Informed Consent Statement:** Not applicable.

**Data Availability Statement:** Not applicable.

**Acknowledgments:** We would like to thank the Balkhash Branch of FishRPC LLP.

**Conflicts of Interest:** The authors declare no conflict of interest. The funders had no role in the design of the study; in the collection, analyses, or interpretation of data; in the writing of the manuscript or in the decision to publish the results.

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
