# Peer review of "Biological Characteristics of a Rare and Vulnerable Species (SCHIZOTHORAX ARGENTATUS (Kessler, 1874)) of TOKYRAUYN RIVER and Approbation of Its Artificial Reproduction"

_agriculture, doi:10.3390/agriculture12081121_

Round 1

Reviewer 1 Report

Dear Authors,

I think it is an important article in terms of endemicity of the species.  Also artificial reproduction of fish is becoming increasingly important. In Table 4,  material does not say anything about how the age is determined in the method. Therefore, it would be appropriate to explain how the age determination is made in the material method. Apart from that, I have shown the necessary corrections on the manuscript.

Best Regards.

Author Response

Point 1: In Table 4, material does not say anything about how the age is determined in the method. Therefore, it would be appropriate to explain how the age determination is made in the material method. Apart from that, I have shown the necessary corrections on the manuscript.

Response 1: Age was determined by annual rings on scales with a microscopic method according to the generally accepted methodology [31-32].

Point 2: It would be better if it was written in lower case, as "vulnerable"

Response 2: wrote in lower letters

Point 3: (Schizothorax argentatus.....) and should be italic.

Response 3: corrected

Point 4: ....of Tokyrauyn River....."

Response 4: corrected

Point 5: quality

Response 5: corrected

Point 6: not in the text; references [33-35]

Response 6: added

Dear reviewer, all your comments have been taken into account and included in the corrected version of the article.

Reviewer 2 Report

The manuscript entitled “Biological characteristics of a rare and vulnerable species (Schizothorax argentatus Kessler, 1874) of to Kyrauyn river and approbation of its artificial reproduction” provides some essential data in the field of agriculture. However, the manuscript writing and the presentation of data need improvement. My major reasons are due to:

1. Too much errors relating to the use of spelling, capital/lowercase letters, italics, units, symbols, decimal points, brackets, and scientific names found throughout the manuscript,

2. In general, almost paragraphs have a disjointed structure, with several abrupt transitions that made reading difficult,

3. The amount of information given about the materials and methods is variable, with insufficient details being given about some of the methods and protocols,

4. Some parts of the results did not match with the given materials and methods, and the presentation is very confusing; it is better to group a similar result under the same heading, and

5. Lack of discussion on the core findings, and the conclusion part did not reflect the major findings.  

Author Response

Point 1: Too much errors relating to the use of spelling, capital/lowercase letters, italics, units, symbols, decimal points, brackets, and scientific names found throughout the manuscript

Response 1: corrected

Point 2: In general, almost paragraphs have a disjointed structure, with several abrupt transitions that made reading difficult

Response 2: The article has been re-read, the transitions have been corrected

Point 3: The amount of information given about the materials and methods is variable, with insufficient details being given about some of the methods and protocols

Response 3: Added details to methods such as hormonal stimulation, fish age determination

Point 4: Some parts of the results did not match with the given materials and methods, and the presentation is very confusing; it is better to group a similar result under the same heading, and

Response 4: grouped under one heading, a sequence of methods and results appeared

Point 5: Lack of discussion on the core findings, and the conclusion part did not reflect the major findings.  

Response 5: Main results by stages of the embryonic period of the life cycle of the Balkhash marinka of development were discussed in more detail.

Dear reviewer, all your comments have been taken into account and included in the corrected version of the article.

Sincerely, Adyrbekova Kamila

Reviewer 3 Report

Dear Authors,

Your article is very well written. It is necessary to do some minor corrections. Below you can find my particular comments:

Cited M&M for hormonal stimulation was conducted on carp. Please, detail this part.

The result part mixed discussion and results partially. It would be better if the result part include only facts, without conclusions.

Line 288 - please add after the figure its number.

It is assumed that in scientific papers in the "results" part, citations are not used - they are related to the discussion.

Table 5 - please add the units in the brackets - in all tables.

Figure 3. Please mark what you are present in both pictures. Please add the letters for both parts of this photo-panel.

Figure 6 - 8 - please add marks where is blastodisc in these photos. Please, explain in the legend what you are presenting in each of these photographs. I suggest you put all of these photographs into one photo panel with a full figure legend.

Sincerely

R.

Author Response

Dear reviewer, thank you for your comments,

Point 1: Cited M&M for hormonal stimulation was conducted on carp. Please, detail this part.

Response 1: a new table was added and the method of hormonal stimulation was described

Hormonal stimulation of spawning of breeders in artificial conditions and natural environment is carried out under recommendations [37-39]. Hormonal stimulation to producers of the Balkhash marinka was carried out with acetonated pituitary glands of cyprinids collected during the period of pre-spawning migration of fish, previously treated with acetone and dried. This method is considered classical and is used in the reproduction of carp, grass carp and silver carp in the conditions of fish farms.

The stage of egg maturation and readiness for spawning was stimulated by pituitary injections. Given the initial state of the females and the low temperature regime in the pools of the fish breeding area (16.6-17.8°C), a three-fold injection scheme with acetonated carp pituitary gland was applied to them. The first (preliminary) dose of the drug to females was 0.3 mg per kg of body weight. The choice of such a small dose ensures the progressive nature of the process of maturation of the eggs with the existing hormonal background. Permissive injection performed after 12 hours. Observation of the state of the injected females showed that the injected dose of the permissive injection was not sufficient to obtain mature reproductive products, and after 36 hours the drug was additionally administered in an amount of 2.0 mg/kg (Table 6).

Table 6. - Timing of hormonal stimulation and recommended doses of carp pituitary solution used to stimulate marinka producers

Venue

Time between injections (hours)

Time of ovulation of mature eggs after the last injection (hours)

Preliminary (mg/kg)

Permissive

(mg/kg)

second permissive (mg/kg)

Tokyrauyn River

12 (0,3)

12 (3,0)

12 (2,0)

11-12

Incubation manufactory of recycling aquaculture system

12 (0,3)

12 (3,0)

12 (2,0)

28-32

Point 2: The result part mixed discussion and results partially. It would be better if the result part include only facts, without conclusions.

Response 2: in view of the fact that it is possible to combine the results and discussion, it was done in this way, since unfortunately such work has not been carried out and it is impossible to give much information for discussion

Point 3: Line 288 - please add after the figure its number.

Response 3: corrected

Point 4: It is assumed that in scientific papers in the "results" part, citations are not used - they are related to the discussion.

Response 4: changed

Point 5: Table 5 - please add the units in the brackets - in all tables.

Response 5: corrected

Point 6: Figure 3. Please mark what you are present in both pictures. Please add the letters for both parts of this photo-panel.

Response 7: corrected

Point 7: Figure 6 - 8 - please add marks where is blastodisc in these photos. Please, explain in the legend what you are presenting in each of these photographs. I suggest you put all of these photographs into one photo panel with a full figure legend.

Response 6: 

Figure - Stages of the embryonic period of the life cycle of the Balkhash marinka;

Ⅰ - Stage III - Activation of the egg and formation of the blastodisc; ⅠⅤ stage - Gastrulation; Stage Ⅴ - Organogenesis; VI -ⅤⅠI stage - Separation of the tail section from the yolk sac; ⅤⅠⅠⅠ stage - Postembryo of the Balkhash marinka, development of the gill-jaw apparatus;

Arrows - 1 - caviar shell; 2 - perivitelline space; 3 - yolk sac; 4 - blastodisk; 5 - neural tube; 6 - somites; 7 - the head section of the embryo; 8 - tail section of the embryo.

Sincerely, Adyrbekova Kamila

Round 2

Reviewer 2 Report

The manuscript was well revised by the authors.